# *IL-6* and *HSPA1A* Gene Polymorphisms May Influence the Levels of the Inflammatory and Oxidative Stress Parameters and Their Response to a Chronic Swimming Training

**DOI:** 10.3390/ijerph19138127

**Published:** 2022-07-02

**Authors:** Jadwiga Kotowska, Ewa Jówko, Igor Cieśliński, Wilhelm Gromisz, Jerzy Sadowski

**Affiliations:** 1Department of Natural Sciences, Faculty of Physical Education and Health in Biała Podlaska, Józef Piłsudski University of Physical Education in Warsaw, 21-500 Biała Podlaska, Poland; jadwiga.kotowska@awf.edu.pl; 2Department of Sports Sciences, Faculty of Physical Education and Health in Biała Podlaska, Józef Piłsudski University of Physical Education in Warsaw, 21-500 Biała Podlaska, Poland; igor.cieslinski@awf.edu.pl (I.C.); wilhelm.gromisz@awf.edu.pl (W.G.); jerzy.sadowski@awf.edu.pl (J.S.)

**Keywords:** gene polymorphism, SNPs, inflammation, prooxidant-antioxidant status, HSP70, TNFα, swimming training, healthy men

## Abstract

The aim of the study was to evaluate whether the most common polymorphisms in the *IL-6* and *HSP70* genes affect the circulating heat shock protein 70 (HSP70), as well as inflammatory and prooxidant-antioxidant parameters in healthy men undergoing chronic endurance training. The subjects were randomly assigned to a 12-week swimming training (ST group) or control group (CON). Fasting blood samples were collected pre- and post-study period to assessment: superoxide dismutase (SOD) and glutathione peroxidase (GPx) activities, serum levels of lipid hydroperoxides (LHs), tumor necrosis factor α (TNFα), and HSP70. Subjects were genotyped for *IL-6*-174G/C, *HSPA1A* + 190 G/C and *HSPA1B* + 1538 A/G single nucleotide polymorphisms (SNPs) by real-time PCR. After a 12-week study period, a decrease in TNFα, HSP70, and GPx was observed in the ST group, but not the CON group. *IL-6* SNP affected serum TNFα levels (main effect of genotype). Higher TNFα levels (pre- and post-study period) was observed in CC CON than in other *IL-6* genotypes of CON and ST groups. However, a post-training decrease in TNFα was observed in both GG and CC *IL-6* genotypes of ST group. In turn, only GG *IL-6* genotype of the ST group was related to a post-training decrease in HSP70 (main time and genotype interaction). Moreover, pre- and post-training LHs were lower in GG than GC/CC *HSPA1A* genotypes of the ST group (main genotype effect). In conclusion, polymorphisms within the *IL-6* and *HSPA1A* genes seem to affect baseline levels of some inflammatory parameters and prooxidant-antioxidant status and/or their changes after chronic swimming training. However, the results should be confirmed in a study with a larger sample size, one that includes individuals with sedentary lifestyles.

## 1. Introduction

It is well known that regular physical activity, apart from enhancing the expression of genes encoding antioxidant enzymes in skeletal muscle cells, also induces a systemic change in many cytokines with inflammatory and anti-inflammatory properties that protect against chronic disorders associated with low-grade systemic inflammation [1,2].

Among the main mediators of inflammation, IL-6 plays a central role in the inflammatory process and in the negative prognosis of some aging-associated diseases, especially in sedentary individuals [3]. It is secreted by monocytes, endothelial cells, and fibroblasts and regulates immune response, the acute phase response, and inflammation [4]. It has been reported that the *IL-6*-174G/C (rs1800795) single nucleotide polymorphism (SNP) functionally affects *IL-6* gene promoter activity, regulating transcriptional process and cytokine levels, and modulating inflammatory phenotype [2]. Indeed, significant differences in circulating IL-6 levels between genotypes within the *IL-6*-174G/C SNP have been reported in various disease states. However, findings regarding which genotype is beneficial or detrimental in terms of inflammatory disease risk are inconclusive. 

Recent systematic reviews with meta-analysis have shown that the CC homozygous genotype is an unfavorable genotype that increases susceptibility to coronary artery disease [5,6]. These findings are consistent with a previous prospective, population-based study (Rotterdam Study) that found greater arterial stiffness as a precursor to cardiovascular disease in individuals with the CC genotype compared to individuals with the wild-type GG genotype [7]. On the contrary, other genetic association studies have reported a higher risk of obesity [8] or dyslipidemia, especially in the postprandial phase [9], with higher promoter activity of the G allele of rs1800795 compared to the C allele. However, in the latter studies [8,9], unlike the former, the G allele was a rare allele in the study populations. Thus, a possible explanation for these contradictions can be formulated through the heterogeneity of the different groups studied (also in terms of ethnicity), as well as other environmental factors. Namely, it should be noted that IL-6 has a pleiotropic effect, depending on the source of origin. As a multifunctional cytokine, muscle-derived IL-6 may play an important role in metabolic signaling, regulating energy expenditure during exercise and promoting muscle recovery through satellite cell activation [3]. Higher serum IL-6 levels, both at baseline and in response to acute exercise, were found in men with -174GG genotype compared to GC/CC genotype carriers [3]. Consistent with this, the GG genotype and G allele were found to be over-represented in Spanish [10] and Polish [11], but not Israeli [12], elite power athletes. In contrast, the C allele was found to be associated with eccentric exercise-induced skeletal muscle damage [13,14]. 

Apart from cytokines, synthesis of the major stress-inducible heat shock protein 70 (Hsp70) is related to inflammatory processes. Heat-shock proteins (HSPs) act as molecular chaperones playing an important role in normal cell processes, and they help in protein folding, assembly, disassembly, and translocation of other proteins. HSPs are classified according to their molecular weight, which includes HSP-70 at 70 kDa. These proteins are associated with cell protection through inhibition of apoptosis [15]. Moreover, HSP70s seem to play a complementary role in protection against oxidative damage, especially because HSP70s can recover oxidatively denatured proteins [16].

Physical exercise and associated physiological alterations induce HSP70 expression in many tissues and cell types, not only in skeletal muscle cells, but also in cardiac muscle cells, liver, and lymphocytes. The breakdown of cellular homeostasis induced by changes in temperature, pH, ion concentration, oxygen partial pressure, glycogen/glucose, and ATP depletion are among the factors that activate HSP70 synthesis during exercise. In addition, free radical production was an important factor involved in the induction of HSP70 in response to exercise, and a state of oxidative stress was simultaneously associated with high intracellular HSP70 concentrations [16].

HSPs, which are released extracellularly (eHSPs), play a key role in the induction of cellular immune responses [17]. As reported by Bekos et al. [18], elevated HSP70 levels in serum and white blood cells can be an indicator of predisposition to exercise-induced bronchoconstriction (EIB) in non-professional non-asthmatic runners. It was also found that serum HSP70 levels may be a useful diagnostic and prognostic factor in patients suffering from various diseases usually associated with oxidative stress and/or inflammation [16], among others rheumatoid arthritis [19], breast cancer [20], diabetes [21], or multiple sclerosis [22].

It has been suggested that genetic variations in the *HSP70* genes may have an influence on eHSP70 protein induction [23]. In humans, the HSP-70 family is encoded by *HSP70-1* (*HSPA1A*), *HSP70-2* (*HSPA1B*) and *HSP70-hom* (*HSPA1L*) genes mapped within the MHC class III region (6p21.3). *HSP70-1* and *HSP70-2* genes encode an identical heat-inducible protein HSP-70, and they differ in their regulatory domains, whereas *HSP70-hom* encodes a non-heat-inducible form [15].

The SNPs in the *HSP-70* gene have been reported in different populations and were positively correlated with the level of circulatory cytokines in response to inflammatory stimuli [15]. The most common *HSPA1A* G + 190C SNP has been found to be a functional polymorphism, with a homozygous +190 CC genotype that increases susceptibility to lung cancer [24], coronary heart disease [25], including acute coronary syndrome [23]. In turn, the *HSPA1B* (A + 1538G) SNP was associated with renal complications in Type 2 Diabetes Mellitus [15].

Current knowledge suggests that HSP70 may have important functions in cytoprotection, immune regulation, regeneration, and adaptive processes during exercise and training. Their diagnostic potential as biomarkers for monitoring exercise and training has also been demonstrated [26]. Although physical activity is well known as a non-pharmacological strategy to enhance antioxidant defenses, resulting in reduced oxidative stress parameters [27], the relationship between eHSP70 levels, cytokines, and prooxidant-antioxidant status has not been previously described. Moreover, apart from the existing evidence of an association between *HSP70* gene SNPs and various disease states, there is a lack of research on the effects of SNPs within *HSP70* genes, along with *IL-6* gene SNPs, on eHSP70, together with inflammation and prooxidant-antioxidant status and their response to exercise/training. 

Therefore, the aim of this study was to evaluate whether chronic endurance training (12-week swimming training) in healthy men induces changes in circulating HSP70, TNFα (as pro-inflammatory cytokine), and prooxidant-antioxidant status (lipid hydroxiperoxides, superoxide dismutase, glutathione peroxidase), and whether these changes are dependent on the most common SNPs in *HSP70* genes, i.e., *HSPA1A* + 190 G/C (rs1043618) and *HSPA1B* + 1538 A/G (rs 2763979) as well as in *IL-6* gene: −174G/C (rs1800795).

## 2. Materials and Methods

### 2.1. Study Population and Design

The study included 63 Polish (Caucasian) male physical education students. Initially, 68 students were recruited and randomly assigned to a 12-week swim training program (ST group, *n* = 38) or a control group (CON group, *n* = 30) in which no training protocol was used. All students completed a 12-week swimming training program and were evaluated at the end of the study period. In contrast, five students in the CON group dropped out of the study (personal reasons). Finally, 25 students from the CON group were included in the analysis. None of the study subjects practiced high-performance sports at the time of enrollment. Exclusion criteria of the study were: the use of tobacco products, alcohol consumption, a history of recent surgery or illness, and intake of medications or dietary supplements during four weeks preceding the study. They were asked to refrain from modification of their diet during the study period. The study was conducted in accordance with the principles of the Declaration of Helsinki. All participants gave their consent to participate in the study and the research protocol was approved by the Local Ethics Committee of the Józef Piłsudski University of Physical Education in Warsaw. 

### 2.2. Training Procedure

The training procedure was the same as previously described by Jówko et al. [27]. The swimming training program lasted 12 weeks, with four sessions per week (Monday, Tuesday, Wednesday and Thursday; at 8.45 p.m.; 48 training sessions in total; 1.5 h/day). The trainings were held at a 25-m pool, and each of them was preceded by a 10- to 20-min warm-up. Mean training volume and intensity were the same for all the study subjects and did not change across the study period. The trainings consisted primarily of aerobic work in front crawl. 

### 2.3. Blood Sampling and Biochemical Analyses

Blood samples from the ulnar vein were obtained in the morning, after an overnight fast, prior to (pre) and after (post) a 12-week study period in the CON group, or prior to and after a 12-week swimming training (48 h after the last training session) in the ST group. Blood was collected to four tubes: three (2 × 1 mL and 1 × 2 mL) with EDTA as an anticoagulant (for isolation of DNA and analyses of whole blood and erythrocytes) and one (6 mL) without an anticoagulant (to separate serum). Two portions of the blood (6 mL and 2 mL) were centrifuged at 3000× *g* for 10 min (4 °C) to separate serum and erythrocytes, respectively. Following the centrifugation, erythrocytes were washed three times with a cold isotonic saline solution. Whole blood, erythrocytes and sera were frozen and stored at −80 °C until analysis.

Serum Hsp70 and TNFα concentration were analyzed using the ELISA (enzyme-linked immunosorbent assay) method in compliance with the manufacturer’s instructions. HSP70 and TNFα concentrations were evaluated using a Biorbyt Human ELISA kits (Cat. No. orb397059 and orb50111, respectively, Biorbyt Ltd., Cambridge, UK). Serum samples for concentration evaluation were not diluted before analysis.

Results were expressed in pictograms per milliliter. The detection range for the kits is between 125–8000 pg/mL and 7.8 pg/mL–500 pg/m for the HSP70 and TNFα, respectively. The mean minimum detectable dose (sensitivity) of the assays is 60 pg/mL for HSP70, and 1 pg/mL for TNFα. All samples’ absorbance was read at 450 nm (ELx808 Absorbance Microplate Reader, BioTek Instruments Inc., Santa Clara, CA, USA).

SOD and GPx activities were determined with commercially available kits (RANSOD Cat. No. SD 125 and RANSEL Cat. No. RS 505, respectively; Randox, Crumlin, UK). The details of the method were described previously by Kotowska J. et al. [28]. LHs were determined with a commercially available kit (LPO-586, OXIS Internatl., Portland, OR, USA).

### 2.4. Genotyping

Genomic DNA for genotyping was isolated from peripheral venous blood using QIAamp DNA Blood Mini Kit (Qiagen GmbH, Hilden, Germany). The concentration of DNA was determined with Picodrop microliter spectrophotometer (PicoDrop, Saffron Walden, UK). The following gene polymorphisms were genotyped: *HSPA1A* (rs1043618), *HSPA1B* (rs2763979), and *IL-6* (rs1800795). Genotyping for all gene polymorphisms was carried out by a 25-µL PCR reaction on DNA (100 ng) using a TaqMan PCR Master Mix (Applied Biosystems, Foster City, CA, USA) and fluorescent 5′-exonuclease TaqMan SNP assays (Applied Biosystems, Foster City, CA, USA) with FAM and VIC fluorophore-labeled probes. Real-time PCR was performed on Rotor Gene (Qiagen GmbH, Hilden, Germany), according to the following protocol: an initial 10 min at 95 °C followed by 45 cycles of 15 s at 95 °C, 60 s at 58 °C for *HSPA1A*, 56 °C for *HSPA1B*, 60 °C for IL-6, 30 s at 72 °C, and finally 8 min at 72 °C. *HSPA1A* (rs1043618), *HSPA1B* (rs2763979), and *IL-6* (rs1800795) were genotyped with a commercially available TaqMan kit (C_11917510_10, C_3052606_10, C_1839697_20 AB assay, respectively, Applied Biosystems, Foster City, CA, USA).

### 2.5. Statistical Analysis

Statistical analysis was conducted with Statistica version 13.3 (StatSoft, Krakow, Poland) with the Bonferroni post-hoc test for multiple comparisons. The normal distribution of all variables was confirmed with the Shapiro–Wilk test and visual inspection (quantile distribution plots). All values were reported as mean ± standard deviation (SD). The level of statistical significance was set at *p* < 0.05. Statistical significance of intergroup (CON vs. ST) differences in anthropometric characteristics and was verified with unpaired Student *t*-test. The data regarding biochemical parameters in the whole groups were analyzed with 2 (groups: CON and ST) × 2 (time points: pre- and post-) factorial (two way) ANOVA, with the Bonferroni post-hoc test for multiple comparisons. Relationships within pairs of the biochemical parameters were analysed on the basis of Pearson’s coefficients of linear correlation.

For each SNP, the deviation of the genotype frequencies from those expected under the Hardy-Weinberg equilibrium was assessed in both groups with chi-square (χ^2^) test [29]. Genotype frequencies in CON and TR were compared with a likelihood ratio (χ^2^ test). 

Variance in biochemical parameters associated with gene polymorphisms were analyzed in both groups (CON and ST) simultaneously by means of 6 or 4 (6 genotypes for *IL-6* gene polymorphism-3 genotypes per group: GG CON, GC CON, CC CON, GG ST, GC ST and CC ST; 4 genotypes for *HSPA1A* polymorphisms: GG CON, GC/CC CON, GG ST and GC/CC ST; 4 genotypes for *HSPA1B* polymorphisms: GG CON, AG/AA CON, GG ST and AG/AA ST- due to very low frequency of homozygote with a rare allele) × 2 (time points: pre and post) two-way factorial ANOVA, with Bonferroni post-hoc test for multiple comparisons. Furthermore, the effect size for the ANOVA was estimated (eta squared; η^2^). Due to the high variability of the biochemical parameter values, all ANOVA analyses were performed on the data after logarithmic transformation (natural logarithm).

## 3. Results

The anthropometric characteristics of participants in the CON and the ST groups are shown in Table 1. No intergroup differences were found in these parameters (*p* > 0.05).

Table 2 presents the level of biochemical parameters in blood. The significant main time effect was seen in serum concentration of TNFα (*p* = 0.005, with medium effect size). This parameter decreased significantly in the ST group after 12-week swimming training (*p* < 0.05), whereas persisted unchanged in the CON group. In case of serum HSP70 levels, a tendency to main effects of time (*p* = 0.07) and group (*p* = 0.06) was observed. Moreover, a significant difference between the ST and the CON groups was seen in post-values of HSP70 (*p* < 0.05), with main effect of time x group interaction (*p* = 0.001, and large effect size). No significant main effects were observed in the level of LHs in blood. On the contrary, the main effect of the group concerned SOD activity (*p* = 0.04, with a medium effect size); however, without any significant intra- or inter-group differences based on Bonferroni post-hoc tests. In turn, a significant decrease in GPx activity was found in the ST group after 12-week swimming training, with significant time and group interaction (*p* = 0.03, and mean effect size). 

For all SNPs in the genes analysed in our study, χ^2^ test confirmed that the observed frequencies did not deviate from the Hardy-Weinberg equilibrium, with no significant inter-group differences in genotype frequencies (Table 3). 

Table 4 presents the results in both the CON and the ST groups, stratified according to *IL-6* SNP. After 12 weeks of study period, serum TNFα decreased (main time effect, *p* = 0.01, with medium effect size); this decrease was significant in both CC and GG genotypes of ST group (*p* < 0.01). Significant differences in serum TNFα, both pre and post 12-week study period were observed between *IL-6* genotypes (main effect of genotype, *p* = 0.04, with large effect size). Namely, before 12-week study period, TNFα was significantly higher in CC genotype carriers from the CON group than in GC genotype carriers from the same group, as well as in GG and GC genotype carriers from the ST group (*p* < 0.05). In turn, after the 12-week study period, CC genotype in CON group showed significantly higher TNFα levels as compared to the other two genotypes in the CON group and all three genotypes in the ST group (*p* < 0.05).

In case of serum level of HSP70, a tendency to main time effect (*p* = 0.09, small effect size) and significant main time and *IL-6* genotype interaction (*p* = 0.045, large effect size) were observed (Table 4). A decrease in HSP70 levels was found in GG genotype in the ST group (*p* < 0.01) after 12-week swimming training, and its post-training level was lower than in the GC and GG genotypes of the CON group (*p* < 0.05; Table 4).

No main effects were observed for both GPx and LHs (Table 4). In turn, for SOD activity, a trend toward a main genotype effect was found (*p* = 0.08, large effect size; Table 4).

The results in both the CON and the ST groups, stratified according to *HSPA1A* SNP are presented in Table 5. In contrast to *IL-6* SNP, *HSPA1A* polymorphism did not affect serum level of TNFα, since only time main effect was observed (*p* = 0.005; medium effect size), although post-training decrease in TNFα was significant only in GC/CC genotypes of the ST group (*p* < 0.05). In case of serum level of HSP70, a tendency to the main effect of time was found (*p* = 0.07). However, after the 12-week study period, a significant decrease in HSP70 was observed in GG and GC/CC genotypes of *HSPA1A* SNP in the ST group with the main effect of time and genotype interaction (*p* = 0.01, large effect size). Moreover, the main effect of genotype was found in LHs results stratified according to *HSPA1A* SNP (*p* = 0.04, medium effect size). In GG *HSPA1A* genotype of the ST group, both pre- and post-training LHs were lower than in GC/CC genotypes of ST (*p* < 0.05). In turn, a tendency for the main effect of genotype was related to SOD activity (*p* = 0.08), whereas no main effects or tendency were found for GPx results stratified according to *HSPA1A* SNP (Table 5). On the contrary to both *IL-6* and *HSPA1A* SNPs, *HSPA1B* SNP did not affect biochemical parameters studied in a significant manner (data not shown).

Table 6 and Table 7 show correlations between biochemical parameters. Taking into account the whole group of students at baseline (prior to the 12-week study period, Table 6), SOD activity was positively correlated with TNFα (r = 0.30, *p* = 0.045) and HSP70 (r = 0.60, *p* = 0.00001). In turn, in ST group after the 12-week swimming training (Table 7), positive correlations were found between HSP70 and TNFα (r = 0.36, *p* = 0.012), as well as between SOD activity and LHs (r = 0.36, *p* = 0.034). Furthermore, GPx activity was positively correlated with both TNFα (r = 0.56, *p* = 0.0004) and HSP70 (r = 0.42, *p* = 0.01). Moreover, SOD activity showed a tendency to be positively correlated with HSP70 (r = 0.31, *p* = 0.07).

## 4. Discussion

The main finding of the current study is that, in healthy male population, the polymorphisms within *IL-6* and *HSPA1* genes affect basal levels of the inflammatory/oxidative stress parameters and/or their changes in response to chronic swimming training.

It has previously been suggested that long-term adaptation to exercise, including anti-inflammatory effects, is mediated by muscle-derived IL-6 [30], which is involved in immune function, muscle repair, and hypertrophy following exercise-induced damage [12]. Exercise can induce increases in muscle derived IL-6 mRNA and subsequent elevations in circulating IL-6 [31,32]. This response in circulating IL-6 following exercise is related to exercise intensity and duration, the mass of recruited muscles, and endurance capacity [32]. IL-6 is thought to play an important role in regulating the immune system, as it can stimulate circulating anti-inflammatory cytokines such as IL-1Ra and IL-10 and inhibit the production of pro-inflammatory TNF-α [30].

An allelic variant within the most common *IL-6* SNP -174G/C has also been shown to affect IL-6 expression and modulate the inflammatory state. In our study, the *IL-6*-174G/C SNP affected circulating TNFα (with significant main effect of genotype; *p* = 0.04, large effect size). Namely, the -174 CC genotype of the CON group was associated with higher serum TNFα levels compared with the other genotypes at both study periods, i.e., before and after the 12-week study period. On the other hand, it should be emphasized that the participants of our study were physical education students. Thus, in healthy young, physically active men, a major regulator of immune function appears to be muscle-derived IL-6 as an anti-inflammatory myokine. We did not measure serum IL-6 concentrations; however, previous studies on the effect of *IL-6*-174G/C SNP on circulating IL-6 showed that the -174 GG genotype is associated with higher (baseline and post-exercise) serum IL-6 concentrations compared with GC and CC genotypes [3]. Given that IL-6 inhibits TNFα production, this explains the lower serum TNFα levels in GG genotype carriers, compared to CC genotype carriers in our study (i.e., GG ST vs. CC CON before study period, as well as GG CON and GG ST vs. CC CON after 12-week period).

Our results may indicate more pronounced pro-inflammatory processes in our participants with the CC genotype than in carriers of at least one G allele. It is consistent with a previous study [33] in which the -174 CC genotype (compared to the GG genotype) was associated with higher susceptibility to upper respiratory tract infections among endurance athletes. In contrast, *IL-6*-174 GG homozygous carrier status has already been linked to reduced susceptibility to acute and chronic inflammatory diseases [5,6,34,35]. On the other hand, the decrease in circulating TNFα after 12-week swimming training in our study, which was observed in the whole ST group, but not in the CON group, was related to both GG and CC genotypes within *IL-6*-174G/C SNP. This may also indicate that the anti-inflammatory effect of chronic swimming training was not related to *IL-6* SNP-174G/C or other factors may influence this response.

It should be noted that the reduction in TNFα levels observed in our study after 12-week training was accompanied by a reduction in serum HSP70 levels. It is thought that acute intense exercise induces HSP expression in various cell types and tissues, and hyperthermia, ischemia, oxidative, cytokine, and muscle stress are potent inducers of HSP expression [36]. It has also been suggested that in addition to their chaperone-like functions, HSP70 have a dual function depending on cellular localization—anti-inflammatory when acting intracellularly and pro-inflammatory when acting extracellularly [16]. Indeed, intracellular HSP70, which has been reported to increase following chronic adaptation to exercise training, plays an anti-inflammatory role. Meanwhile, eHSP70 binding to toll-like receptors provides a pro-inflammatory stimulus that may result in a fatigue signal to the CNS during higher-load exercise sessions, both acute and chronic [16,37]. An earlier study [36] reported higher serum HSP70 levels in athletes (soccer players) than in non-athletes, but the authors did not state during what training period the study was conducted. Thus, it is possible that the reason for the increased eHSP70 concentration in soccer players compared to the control group could be related to the intense training/competition period. Our results indicate that a chronic physical training regimen decreases serum HSP70 levels, which is consistent with the results of Fazzi Gómez et al. [38], who observed a reduction in eHSP70 levels with increasing physical activity habits. Our findings also support the latest results of Lovas et al. [39], who found that eHsp70 concentrations (both at baseline and after acute exercise) are significantly lower in athletes compared to control subjects.

In the current study, there was a significant positive correlation between serum HSP70 and TNFα levels after 12-week swimming training (i.e., in the ST group), which may confirm the association of eHSP70 with inflammation. The results of our study are consistent with a previous study [40] conducted on young tennis players during a 14-day conditioning camp. In that study, both Hsp70 and TNFα serum levels were the highest after the tournament season (preceding the conditioning camp), and then the levels of both parameters decreased after 3 days of active recovery and at the end of the 14-day conditioning camp. Furthermore, during the 14-day camp, serum HSP70 levels were positively correlated with TNFα levels [40]. It is worth mentioning that in our study, within the *IL-6*-174G/C SNP, a significant reduction in eHSP70 occurred only for the GG genotype of ST group (with significant interaction of time and genotype (*p* = 0.045, and large effect size). This may confirm the results of previous studies indicating that there are close interactions between activation of HSP gene expression and IL-6 production. Given that the -174GG genotype has previously been associated with higher blood IL-6 levels [3], our results may indicate an important role for muscle-derived IL-6 in mediating the beneficial effects of endurance training on the body. However, somewhat surprisingly in our study, the 12-week swimming training in ST group resulted in a decrease in GPx activity (with main time and group interaction, *p* = 0.03, and mean effect size). Furthermore, among three *IL-6* genotypes in the ST group, the -174GG genotype showed the most pronounced reduction in GPx activity after the 12-week training (with a trend toward an interaction of time and genotype, *p* = 0.054, based on raw data, before logarithmic transformation; statistic analyses of main effects before transformation are not shown in the tables).

It is believed that reactive oxygen species formed during exercise may act as signaling molecules that increase the expression of antioxidant enzymes. Therefore, overexpression of SOD and GPx genes is considered a positive adaptive response to exercise training in the prevention of cardiovascular diseases [41]. On the other hand, it has also been suggested that antioxidant enzyme activity changes in proportion to exercise-induced reactive oxygen species synthesis [42]. In line with the above, an increase in antioxidant protection in blood may reflect the increased production of free radicals/reactive oxygen species, especially when the activity of antioxidant enzymes is marked in cells which do not have nucleus (erythrocytes) and in extracellular environment (serum, plasma, whole blood). It seems justified to claim that changes in SOD or GPx activities in these environments may only result from enzymatic protein post-translational modifications [43], and not from the increased expression of genes and enzymatic protein synthesis resulting from exercise training. Therefore, we cannot exclude the possibility that the decrease in GPx activity in our study, most pronounced in *IL-6*-174GG genotype, without changes in other redox parameters, may indicate a decrease in production of some reactive oxygen species. This result corresponds to the results of our previous study on wrestlers, where, prior to the training program (at baseline), lower levels of LHs were correlated with lower whole blood GPx activity [44]. Although overall GPx can be considered protective against oxidative stress, it is important to consider that changes in redox balance in either direction, oxidative or reductive, may also influence the protective or harmful roles of GPx. Under certain circumstances, increased GPx expression may promote reductive stress by scavenging essential oxidants, leading to deleterious physiological effects such as impairment of growth factor-mediated signalling, important in many adaptive mechanisms [45].

In our study, in opposition to GPx activity, SOD activity was not significantly changed in the whole ST group after 12 weeks of swimming training. However, we found significant correlations of prooxidant-antioxidant balance parameters with HSP70 and TNFα. Namely, in the whole study population (*n* = 63), at the beginning of the study, a significant positive correlation was found between SOD and eHSP70, as well as between SOD and TNFα. In turn, after 12 weeks of swimming training in the ST group (*n* = 38), GPx activity was positively (significantly) correlated with both eHSP70 and TNFα. There was also a trend toward a positive correlation between SOD and eHSP70, whereas a significant positive correlation was found between SOD and LHs. All these results mentioned above support the hypothesis of Margaritis et al. [42] that in healthy individuals, the current state of free radical/reactive oxygen species production is a factor modulating the activity of antioxidant enzymes in the blood. Therefore, consistent with the above, lower levels of inflammatory markers such as eHSP70 and TNFα appear to be accompanied by lower levels of oxidative stress markers (lower levels of LHs as products of lipid peroxidation by free radicals) and lower antioxidant enzyme activities.

Our finding that *IL-6*-174GG genotype may be related to diminished levels of some inflammatory/oxidative stress parameters (as compared to other genotypes) is in line with the results of the study carried by Yamin et al. [14] who found a strong association of the *IL-6* G-174C genotype with systemic CK response to strenuous exercise, which is related to muscle damage induced by many factors, among other oxidative stress and inflammation. Namely, they observed that carriers of the *IL-6*-174 CC genotype, as compared to GG genotype, showed a steeper increase and reached higher maximal CK values following eccentric arm exercise. According to the Authors, this increase in CK in the CC genotype (with reduced systemic IL-6 levels) supports the notion that, during exercise, IL-6 functions as a myokine and serves to reduce muscular damage in response to exercise, and that athletes who are able to produce more IL-6 during exercise may recover faster from a similar training levels due to reduced inflammatory response. However, our findings have to be confirmed in further studies with a more numerous group of subjects, both men and women, as well as subjects who are not physically active. All these issues should be considered as limitations of the current study. Moreover, more detailed analysis of multiple biochemical parameters, including serum IL-6 measurements are needed to confirm our results, especially given the results of another study [46] involving military recruits undergoing 8 weeks of basic training. In that study, *IL-6*-174 CG was the only phenotype that showed significantly higher plasma IL-6 levels after acute aerobic exercise, with no significant intergenotypic differences in IL-6 response to 8-week military training [46].

In comparison to the *IL-6* gene polymorphism, the *HSP70* gene polymorphisms, especially *HSPA1B* SNP, had considerably minor effects on the parameters examined in our study. It should be noted, however, that we considered a dominant model for the analysis (wild-type homozygote versus heterozygote together with homozygote mutant), which was dictated by the low frequency of mutant homozygous in the study population (only 3 and 2 individuals in the ST and in the CON group, respectively). In case of *HSPA1B* SNP, only the main time effect was observed in TNFα levels (data not shown). Similarly, the *HSPA1A* SNP seemed to have no effect on the TNFα results. The G + 190C *HSPA1A* polymorphism appears to play a functional role in affecting Hsp70 protein synthesis and eHsp70 levels [23]. The +190C allele was found to be more sensitive to stressful stimuli because, compared with the other two genotypes, carriers of the homozygous +190CC genotype had higher baseline plasma Hsp70 levels and were predisposed to develop acute coronary syndrome [23]. In addition to medical studies, Kresfelder et al. [47] found the possibility of using eHSP70 levels and *HSP70* gene polymorphisms as markers of acclimatization. In that study, acclimated subjects showed reduced baseline eHsp70 levels and increased ability to induce eHsp70 with a specific combination of *HSP70* genotypes [47].

In our study, the main effect of time and *HSPA1A* genotype interaction was found for serum HSP70 levels (*p* = 0.01, with large effect size). However, the decrease in HSP70 levels occurred in both GG and GC/CC genotype carriers of ST group after the 12-week training. Moreover, no significant main effects of genotype and interaction between time and genotype were found for both GPx and SOD activities. On the other hand, contrary to *IL-6* SNP, the G + 190C *HSPA1A* SNP affected serum LHs (main effect of genotype, *p* = 0.04, mean effect size). Both at baseline and after 12-week training, significantly lower concentration of LHs in serum was observed in subjects with the GG genotype in the ST group compared with those with the GC/CC genotypes of the ST group. Our results may indicate that the mutant C allele within the G + 190C *HSPA1A* SNP is associated with elevated levels of oxidative stress, which is consistent with previous findings that the +190CC genotype is more susceptible to stress stimuli [23].

## 5. Conclusions

In a population of healthy men, a 12-week swimming training seems to attenuate inflammatory processes as evidenced by a reduction in circulating TNFα and HSP70 levels. These inflammatory parameters correlated positively with lipid peroxidation and antioxidant enzyme activity. Polymorphisms within the *IL-6* and *HSPA1A* genes affect baseline levels of inflammatory parameters and prooxidant-antioxidant status and/or their changes in response to 12-week swimming training. Primarily, homozygous wild-type genotypes of both *IL-6* (GG) and *HSPA1A* (GG) appear to be associated with lower baseline levels of inflammatory/oxidative stress markers, as well as with their decline after chronic swimming training. However, the results should be confirmed in a study with a larger sample size, on that includes both men and women as well as individuals with sedentary lifestyles.

## Figures and Tables

**Table 1 ijerph-19-08127-t001:** Anthropometric characteristics of the control (CON) and swimming training group (ST) prior to (pre) 12-week study period.

	CON (*n* = 25)	ST (*n* = 38)
Age (years)	21.3 ± 1.3	21.1 ± 1.2
Height (cm)	179 ± 5.0	180 ± 6.0
Body mass (kg)	78.0 ± 6.6	79.3 ± 7.2

There are no significant differences between groups in anthropometric parameters.

**Table 2 ijerph-19-08127-t002:** Biochemical parameters in blood prior to (pre) and after (post) 12-week in the study group (control, CON; swimming training, ST).

Blood Parameter	Time	CON Group (*n* = 25)	ST Group (*n* = 38)	Main Effects: *p*-Values; η^2^ (Effect Size) after Logarithmic Transformation		
				Time	Group	Time × Group
TNFα [pg/mL]	Pre	25.52 ± 0.42	25.41 ± 0.40	*p* = 0.005	*p* = 0.16	*p* = 0.39
	Post	25.37 ± 0.57	25.13 ± 0.26 *	η^2^ = 0.12	η^2^ = 0.02	η^2^ = 0.01
HSP70 [pg/mL]	Pre	570.23 ± 236.22	523.23 ± 314.16	*p* = 0.07	*p* = 0.06	*p* = 0.001
	Post	602.89 ± 256.31	375.01 ± 268.51 *^,a^	η^2^ = 0.05	η^2^ = 0.08	η^2^ = 0.17
SOD [U/g Hb]	Pre	1638.63 ± 147.58	1509.25 ± 180.72	*p* = 0.75	*p* = 0.04	*p* = 0.32
	Post	1563.46 ± 98.79	1523.20 ± 168.79	η^2^ = 0.002	η^2^ = 0.08	η^2^ = 0.02
GPx [U/g Hb]	Pre	29.36 ± 16.83	46.98 ± 25.85	*p* = 0.11	*p* = 0.71	*p* = 0.03
	Post	34.46 ± 24.55	33.38 ± 25.51 *	η^2^ = 0.04	η^2^ = 0.002	η^2^ = 0.08
LHs [mmol/L]	Pre	3.39 ± 1.45	3.63 ± 1.62	*p* = 0.44	*p* = 0.54	*p* = 0.31
	Post	3.35 ± 1.20	3.37 ± 1.75	η^2^ = 0.009	η^2^ = 0.006	η^2^ = 0.02

Values are means ± SD; * significant difference (*p* < 0.05) between pre- and post- values within the same study group. ^a^ significant difference between ST and CON groups in post- values of TNFα (*p* < 0.05). Abbreviations: TNFα—tumor necrosis factor α; HSP70—heat shock protein 70; SOD—superoxide dismutase; GPx—glutathione peroxidase; LHs—lipid hydroperoxides; η^2^—eta squared (effect size).

**Table 3 ijerph-19-08127-t003:** The distribution of genotypes for gene polymorphisms in control (CON) and swimming training (ST) group (*n*; (%)).

Gene Polymorphism	CON (*n* = 25)	ST (*n* = 38)	ST vs. CON
*IL-6*			χ^2^ = 0.89 *p* = 0.22
GG	8 (32)	14 (37)	
GC	12 (48)	16 (42)	
CC	5 (20)	8 (21)	
HWE	*p* = 0.90	*p* = 0.70	
*HSPA1A*			χ^2^ = 0.003 *p* = 0.99
GG	12 (48)	18 (47)	
GC	11(44)	17 (45)	
CC	2 (8)	3 (8)	
HWE	*p* = 0.81	*p* = 0.71	
*HSPA1B*			χ^2^ = 0.025 *p* = 0.98
GG	12 (48)	19 (50)	
AG	11 (44)	16 (42)	
AA	2 (8)	3 (8)	
HWE	*p* = 0.81	*p* = 0.88	

HWE—Hardy-Weinberg equilibrium. χ^2^ chi-square (between groups ST vs. CON), *p* values.

**Table 4 ijerph-19-08127-t004:** Biochemical parameters in blood prior to (pre) and after (post) 12-week in the control (CON) and swimming training (ST) group; the results stratified according to *IL-6* gene polymorphism.

Biochemical Parameter	Time Point	CON (*n* = 25)			ST (*n* = 38)		
		GG (*n* = 8)	GC (*n* = 12)	CC (*n* = 5)	GG (*n* = 14)	GC (*n* = 16)	CC (*n* = 8)
TNFα [pg/mL]	pre	25.4 ± 0.3	25.4 ± 0.3	26.1 ± 0.7 ^a^	25.4 ± 0.5	25.3 ± 0.2	25.7 ± 0.4
	post	25.2 ± 0.4	25.1 ± 0.2	26.2 ± 1.1 ^b^	25.1 ± 0.2 **	25.2 ± 0.4	25.2 ± 0.2 **
^†^ Main effects (*p* values; η^2^): Time Genotype Time × genotype		*p* = 0.01; η^2^ = 0.11 *p* = 0.04; η^2^ = 0.17 *p* = 0.47; η^2^ = 0.07					
HSP70 [pg/mL]	pre	646 ± 200	553 ± 149	601 ± 323	578 ± 329	571 ± 377	411 ± 232
	post	626 ± 134	608 ± 287	606 ± 334	307 ± 124 ^c,^**	471 ± 389	348 ± 294
^†^ Main effects (*p* values; η^2^): Time Genotype Time × genotype		*p* = 0.09; η^2^ = 0.05 *p* = 0.14; η^2^ = 0.13 *p* = 0.045; η^2^ = 0.18					
SOD [U/g Hb]	pre	1563 ± 46	1595 ± 150	1755 ± 104	1537 ± 195	1474 ± 200	1522 ± 110
	post	1503 ± 25	1587 ± 123	1580 ± 111	1521 ± 219	1510 ± 121	1552 ± 153
^†^ Main effects (*p* values; η^2^): Time Genotype Time × genotype		*p* = 0.70; η^2^ = 0.003 *p* = 0.08; η^2^ = 0.15 *p* = 0.81; η^2^ = 0.04					
GPx [U/g Hb]	pre	26 ± 19	27 ± 15	35 ± 17	49 ± 28	44 ± 23	50 ± 29
	post	33 ± 29	34 ± 22	37 ± 23	22 ± 14	40 ± 30	41 ± 27
^†^ Main effects (*p* values; η^2^): Time Genotype Time × genotype		*p* = 0.10; η^2^ = 0.05 *p* = 0.73; η^2^ = 0.05 *p* = 0.17; η^2^ = 0.12					
^†^ LHs [mmol/L]	pre	3.6 ± 1.4	3.6 ± 1.5	3.0 ± 1.4	4.1± 1.6	3.5 ± 1.4	3.3 ± 1.8
	post	3.7 ± 1.1	3.6 ± 1.0	2.8 ± 1.5	3.3 ± 2.2	3.5 ± 1.6	3.2 ± 1.4
Main effects (*p* values; η^2^): Time Genotype Time × genotype		*p* = 0.52; η^2^ = 0.007 *p* = 0.83; η^2^ = 0.04 *p* = 0.43; η^2^ = 0.08					

Values are means ± SD; Abbreviations: TNFα—tumor necrosis factor α; HSP70—heat shock protein 70; SOD—superoxide dismutase; GPx—glutathione peroxidase; LHs—lipid hydroperoxides; η^2^—eta squared (effect size). ^†^ Main effects (*p* values, η^2^) and post-hoc regard values after logarithmic transformation (natural logarithm). ** significant difference (*p* < 0.01) between pre- and post-values within the same genotype of ST group. ^a–c^ significant differences between IL-6 genotypes of CON and ST groups: ^a^ pre-values of TNFα: CC CON vs. GC CON, GG ST and GC ST (*p* < 0.05). ^b^ post- values of TNFα: CC CON vs. GC CON, GG CON, CC ST, GC ST and GG ST (*p* < 0.05). ^c^ post-values of HSP70: GG ST vs. GC CON and GG CON (*p* < 0.05).

**Table 5 ijerph-19-08127-t005:** Biochemical parameters in blood prior to (pre) and after (post) 12-week in the control (CON) and swimming training (ST) group; the results stratified according to *HSPA1A* gene polymorphism.

Biochemical Parameter	Time Point	CON (*n* = 25)		ST (*n* = 38)	
		GG (*n* = 12)	GC/CC (*n* = 13)	GG (*n* = 18)	GC/CC (*n* = 20)
TNFα [pg/mL]	pre	25.4 ± 0.4	25.7 ± 0.5	25.4 ± 0.3	25.4 ± 0.5
	post	25.2 ± 0.4	25.6 ± 0.8	25.2 ± 0.3	25.1 ± 0.2 *
† Main effects (*p* values; η^2^): Time Genotype Time × genotype		*p* = 0.005; η^2^ = 0.13 *p* = 0.49; η^2^ = 0.04 *p* = 0.67; η^2^ = 0.03			
HSP70 [pg/mL]	pre	469 ± 177	671 ± 262	539 ± 333	509 ± 287
	post	608 ± 278 ^a^	598 ± 257	410 ± 354 *	344 ± 161 *
† Main effects (*p* values; η^2^): Time Genotype Time × genotype		*p* = 0.07; η^2^ = 0.05 *p* = 0.16; η^2^ = 0.08 *p* = 0.01; η^2^ = 0.18			
SOD [U/g Hb]	pre	1630 ± 92	1647 ± 205	1497 ± 138	1520 ± 216
	post	1599 ± 102	1528 ± 95	1476 ± 123	1566 ± 195
† Main effects (*p* values; η^2^): Time Genotype Time × genotype		*p* = 0.72; η^2^ = 0.002 *p* = 0.09; η^2^ = 0.11 *p* = 0.50; η^2^ = 0.04			
GPx [U/g Hb]	pre	29 ± 15	30 ± 19	43 ± 21	50 ± 30
	post	39 ± 31	30 ± 20	34 ± 27	33 ± 25
† Main effects (*p* values; η^2^): Time Genotype Time × genotype		*p* = 0.12; η^2^ = 0.04 *p* = 0.91; η^2^ = 0.01 *p* = 0.15; η^2^ = 0.09			
LHs [mmol/L]	pre	3.1 ± 0.7	3.7 ± 2.2	3.0 ± 1.5 ^b^	4.2 ± 1.7
	post	3.2 ± 1.5	3.5 ± 0.9	2.9 ± 1.8 ^b^	3.9 ± 1.7
† Main effects (*p* values; η^2^): Time Genotype Time × genotype		*p* = 0.44; η^2^ = 0.01 *p* = 0.04; η^2^ = 0.13 *p* = 0.71; η^2^ = 0.02			

Values are means ± SD; Abbreviations: TNFα—tumor necrosis factor α; HSP70—heat shock protein 70; SOD—superoxide dismutase; GPx—glutathione peroxidase; LHs—lipid hydroperoxides; η^2^—eta squared (effect size). † Main effects (*p* values, η^2^) and post-hoc regard values after logarithmic transformation (natural logarithm). * difference (*p* < 0.05) between pre- and post-values within the same genotype of ST group. ^a^ differences between *HSPA1A* genotypes in post-values of HSP70: GG CON vs. GG ST and GC/CC ST (*p* < 0.05). ^b^ differences between *HSPA1A* genotypes in pre and post-values of LHs: GG ST vs. GC/CC ST (*p* < 0.05).

**Table 6 ijerph-19-08127-t006:** Statistically significant correlations between biochemical parameters analysed prior to (pre) 12-week swimming training in the whole group of students (*n* = 63).

	TNFα	HSP70	SOD	GPx	LHs
TNFα	—	—	r = 0.30 *p* = 0.045	—	—
HSP70	—	—	r = 0.60 *p* = 0.00001	—	—
SOD	r = 0.30 *p* = 0.045	r = 0.60 *p* = 0.00001	—	—	—
GPx	—	—	—	—	—
LHs	—	—	—	—	—

Abbreviations: TNFα—tumor necrosis factor α; HSP70—heat shock protein 70; SOD—superoxide dismutase; GPx—glutathione peroxidase; LHs—lipid hydroperoxides.

**Table 7 ijerph-19-08127-t007:** Statistically significant correlations between biochemical parameters analysed post 12-week month swimming training in the swimming group (*n* = 38).

	TNFα	HSP70	SOD	GPx	LHs
TNFα	—	r = 0.36 *p* = 0.012	—	r = 0.56 *p* = 0.0004	—
HSP70	r = 0.36 *p* = 0.012	—	r = 0.31 *p* = 0.07	r = 0.42 *p* = 0.01	—
SOD	—	r = 0.31 *p* = 0.07	—	—	r = 0.36 *p* = 0.034
GPx	r = 0.56 *p* = 0.0004	r = 0.42 *p* = 0.01	—	—	—
LHs	—	—	r = 0.36 *p* = 0.034	—	—

Abbreviations: TNFα—tumor necrosis factor α; HSP70—heat shock protein 70; SOD—superoxide. dismutase; GPx—glutathione peroxidase; LHs—lipid hydroperoxides.

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
