# Peer review of "IL-6* and *HSPA1A* Gene Polymorphisms May Influence the Levels of the Inflammatory and Oxidative Stress Parameters and Their Response to a Chronic Swimming Training"

_ijerph, 2022, doi:10.3390/ijerph19138127_

Round 1

Reviewer 1 Report

In this manuscript, the authors describes " IL-6 and HSPA1A gene polymorphisms may influence the levels of the inflammatory and oxidative stress parameters and their response to a chronic swimming training". Although this manuscript is generally well written, it would be helpful if the authors address the following concerns:

1) The pool of subjects for this study seem relatively small to make such meaningful correlations and findings for this study. The authors should consider increasing the number of subjects for this study.

2) It would be helpful to readers if the study included both genders.

3) Again, this study would benefit a wider audience if the number of subjects used in this study is not only increased but broadened to include subjects from different races.

4) To reduce unnecessary distractions, table 3 in line 227 should be moved to the next page.

5) In line 243, the footnote for table 4 is not only confusing but also inaccurate. The authors should consider removing the word "table" and fix the spacing as well.

6) Again in line 308, the authors should move table 6 to the next page to reduce distractions.

7) Finally, the study appear to focus on subjects who are healthy, young and physically active. It would be interesting to know of the findings regarding subjects who are not physically active.

In summary, this manuscript would be of benefit to its target audience if the above concerns/issues are addressed.

Reviewer 2 Report

Dear authors, 

The manuscript is interesting and with novel data. However, it is risky to conclude that "In a population of healthy men, 12-week swimming training attenuates inflammation as evidenced by a reduction in circulating TNFα and HSP70 levels"... based on results with statistical differences of around 5% (Tables 2,4,5). Currently, it is recommended that differences in clinical studies be supported by their clinical significance and not just statistical significance. Therefore, discuss these differences based on the percentages, effects size, and R2. With this, to measure the power of the data and results.

Reviewer 3 Report

In this manuscript, a correlation between IL-6 and HP70 genes polimorphisms with inflammatory and oxidative stress parameters after chronic swimming training is performed. Besides the potential correlation that might exist, the manuscript presents some data that are not clear and might have pitfalls that deserve to be fixed before publication.

1)    1) Data organization is not clear and should be changed in order to be able to compare each parameter between the polymorphisms presented in both study groups: control and chronic swimming training.

2)      2) After comparison of some of the data associated with pre-treatment between groups, the authors should realize that there is a huge variability in some of the measurements and a higher n value might be required in some cases. Some examples are reported here.

Table 2: GPx [U/g Hb]  Pretreatment- CON group 29.36 ± 16.83 and ST 46.98 ± 25.85

Table 4 and 5:

                                                             Control  (pretreatment)                                Swimming group (pretreatment)                                                      

                                                        IL6                                                                                IL6

                                 GG (n=8)          GC (n=12)           CC (n=5)                  GG (n=14)        GC (n=16)        CC (n=8)

 HSP70 [pg/ml]     645.6 ± 200.1 552.7 ± 149.3 600.9 ±  322.5                 578.3 ± 329.0  571.3 ± 377.3  410.9 ±231.5 231.5

Although it is clear that the trend  exist in in HSP70 in all samples measured after treatment, the significant change was only observed in the GC group which value drops to 306.5 ± 123.6**. These results might be not completely real due to a low n for the rest of the groups.

 This is an example but the same account for other parameters shown in table 4 and 5.

3)      3) There is huge variability in two parameters (HSP70 and GpX). Please check the values and also evaluate normalizations. Normalizations should be performed as reported using standards but also the amount of protein or volume of plasma that was used in the experiments.

4)      4) Gpx activities and also Units for SOD and Gpx activities seem to be odd [U/g Hb]. Please standardize the data to standard units for these activities (U/dL) of serum.

5)      5) Tables should be shortened to not include values with decimals for a better organization of these values.

6)      6) The title should be changed to show the specific IL-6 polymorphisms associated with the correlation or the generically indicate that the correlation is between IL-6 and HSP70 polymorphisms to keep the coherence.

Round 2

Reviewer 2 Report

Thanks to the authors for efficiently responding to comments.

The manuscript is interesting and now well written, with sufficient background to understand the problem. The results tables and figures are well explained and discussed. They apply validated reliable and sensitive methods. The bibliography is updated.

Reviewer 3 Report

The author's answered all my questions.